# Systematic Review and Meta-Analysis of Particle Beam Therapy versus Photon Radiotherapy for Skull Base Chordoma: TRP-Chordoma 2024

**DOI:** 10.3390/cancers16142569

**Published:** 2024-07-17

**Authors:** Takashi Saito, Masashi Mizumoto, Yoshiko Oshiro, Shosei Shimizu, Yinuo Li, Masatoshi Nakamura, Sho Hosaka, Kei Nakai, Takashi Iizumi, Masako Inaba, Hiroko Fukushima, Ryoko Suzuki, Kazushi Maruo, Hideyuki Sakurai

**Affiliations:** 1Department of Radiation Oncology, University of Tsukuba, Tsukuba 305-8576, Ibaraki, Japan; saitoh@pmrc.tsukuba.ac.jp (T.S.); yli@pmrc.tsukuba.ac.jp (Y.L.); nakamura@pmrc.tsukuba.ac.jp (M.N.); knakai@pmrc.tsukuba.ac.jp (K.N.); iizumi@pmrc.tsukuba.ac.jp (T.I.); hsakurai@pmrc.tsukuba.ac.jp (H.S.); 2Department of Radiation Oncology, Tsukuba Medical Center Hospital, Tsukuba 305-8558, Ibaraki, Japan; ooyoshiko@hotmail.com; 3Department of Pediatric Radiation Therapy Center/Pediatric Proton Beam Therapy Center, Hebei Yizhou Cancer Hospital, Zhuozhou 072750, China; shimizu@pmrc.tsukuba.ac.jp; 4Department of Pediatrics, University of Tsukuba Hospital, Tsukuba 305-8575, Ibaraki, Japan; shohosaka@md.tsukuba.ac.jp (S.H.); minaba-tuk@md.tsukuba.ac.jp (M.I.); fkhiroko@md.tsukuba.ac.jp (H.F.); ryokosuzuki@md.tsukuba.ac.jp (R.S.); 5Department of Child Health, Institute of Medicine, University of Tsukuba, Tsukuba 305-8575, Ibaraki, Japan; 6Department of Biostatistics, Institute of Medicine, University of Tsukuba, Tsukuba 305-8575, Ibaraki, Japan; maruo@md.tsukuba.ac.jp

**Keywords:** chordoma, particle beam therapy, systematic review, meta-analysis, TRP

## Abstract

**Simple Summary:**

Chordoma is a rare cancer that often occurs at the base of the skull. Treating skull base chordoma is challenging because the tumor is difficult to completely remove with surgery and has low radiosensitivity. This study compared two types of radiation modality: particle beam therapy (PT) and photon radiotherapy (RT). We found that PT provides better progression-free survival compared to photon RT. However, PT also has a higher risk of causing brain necrosis. Our findings suggest that PT is more effective for controlling skull base chordoma, but careful planning is needed to minimize side effects.

**Abstract:**

[Objective] The aim of this study was to compare the efficacy of particle beam therapy (PT) with photon radiotherapy (RT) for treatment of skull base chordoma. [Methods] A systematic review was conducted for skull base chordoma treated with PT or photon RT reported from 1990 to 2022. Data were extracted for overall survival (OS) and progression-free survival (PFS), late adverse events, age, gender, gross total resection (GTR) rates, tumor volume, total irradiation dose, and treatment modality. Random-effects meta-regression analysis with the treatment modality as an explanatory variable was performed for each outcome to compare the modalities. [Results] A meta-analysis of 30 selected articles found 3- and 5-year OS rates for PT vs. photon RT or combined photon RT/proton beam therapy (PBT) of 90.8% (95% CI: 87.4–93.3%) vs. 89.5% (95% CI: 83.0–93.6%), *p* = 0.6543; 80.0% (95% CI: 75.7–83.6%) vs. 89.5% (95% CI: 83.0–93.6%), *p* = 0.6787. The 5-year PFS rates for PT vs. photon RT or photon RT/PBT were 67.8% (95% CI: 56.5–76.7%) vs. 40.2% (95% CI: 31.6–48.7%), *p* = 0.0004. A random-effects model revealed that the treatment modality (PT vs. photon RT or photon RT/PBT) was not a significant factor for 3-year OS (*p* = 0.42) and 5-year OS (*p* = 0.11), but was a significant factor for 5-year PFS (*p* < 0.0001). The rates of brain necrosis were 8–50% after PT and 0–4% after photon RT or photon RT/PBT. [Conclusion] This study shows that PT results in higher PFS compared to photon RT for skull base chordoma, but that there is a tendency for a higher incidence of brain necrosis with PT. Publication and analysis of further studies is needed to validate these findings.

## 1. Introduction

Chordoma is a rare disease, with an incidence of 0.18 to 0.84 per million people [1]. The common sites of occurrence are the sacrum and skull base, followed by the spine [1]. NCCN guidelines recommend surgical resection as standard treatment, with postoperative radiation recommended for cases in which residual disease is suspected [2]. However, treatment of chordoma presents significant challenges because complete resection is difficult in common regions of the disease, such as the sacrum and skull base, and the tumor is resistant to radiotherapy (RT) and chemotherapy, which complicates the achievement of local control [3,4]. The prognosis for cases with complete resection is significantly better compared to those with incomplete resection [5,6]. Postoperative RT is useful for skull base chordoma, but a high dose of over 65 Gy is required for local control because of the low radiosensitivity [6]. Thus, advanced irradiation techniques are needed to deliver high doses to the tumor while sparing critical organs such as the brainstem and optic nerves.

Particle beam therapy (PT) is frequently used for skull base chordoma due to its high dose concentration [7]. However, while meta-analyses have shown superiority of PT over photon RT for chordomas in general, no meta-analysis has compared PT to photon RT for skull base chordoma [8]. Compared to chordomas in the sacrum or spine, skull base chordomas are characterized by a younger age of onset, difficulty achieving complete resection due to surrounding anatomical structures, and a greater need for postoperative RT [9,10]. Furthermore, skull base chordomas are often treated similarly to skull base chondrosarcomas. However, chondrosarcomas have higher radiosensitivity and clearly superior survival and local control rates compared to chordomas [11]. While several meta-analyses have compared PT and photon RT for skull base chordomas and chondrosarcomas, it is unclear if these results can be directly extrapolated to skull base chordomas [12,13]. To address these issues, we extracted literature on treatment outcomes and background factors for skull base chordomas to compare the therapeutic effects of PT and photon RT.

## 2. Materials and Methods

### 2.1. Selection Criteria for Meta-Analysis

The review was conducted in accordance with the principles and recommendations of the Preferred Reporting Items for Systematic Reviews and Meta-Analyses (PRISMA) [14]. The study has not been registered. PubMed was searched using the keywords “(chordoma OR chondrosarcoma) AND (radiotherapy OR proton OR carbon) AND (skull OR head)” for articles published from 1990 to 2022. Only articles written in English were included. Two reviewers independently screened all retrieved papers. The process for selecting studies for analysis was as follows: (1) Clinical studies related to chordomas or chondrosarcomas located at the skull base were identified. (2) Articles that reported overall survival (OS) or progression-free survival (PFS) were selected. (3) Studies with ten or more cases per treatment modality were included to ensure robust sample sizes for analysis. (4) Studies in which a single treatment modality accounted for at least 80% of the cases were included. (5) For multiple publications from the same institution covering overlapping periods, only the most recent study was included. Data were extracted for the number of cases, 3- or 5-year OS, 3- or 5-year PFS, late adverse events, age, gender, gross total resection (GTR) rates, tumor volume, total irradiation dose, and treatment modality (PT vs. photon RT or combined photon RT/proton beam therapy (PBT)). If the 3- or 5-year OS and PFS were not specified in the text, these rates were estimated from figures.

### 2.2. Statistical Analysis

Random effects meta-analyses of 3- and 5-year OS and PFS were performed for each modality, and forest plots were drawn. For studies with missing accuracy data, missing values were imputed using the number of cases, risk set size at each year, and mean dropout rate. Heterogeneity in each meta-analysis was evaluated by I-square statistics. Random-effects meta-regression with modality as an explanatory variable was performed for each outcome to compare the modalities. All analyses were performed using R version 4.3.2 (R Core Team, Vienna, Austria) and its accompanying meta package [15].

## 3. Results

The selection process and outcomes of the articles are shown in Figure 1. Ultimately, 30 articles met our inclusion criteria [16,17,18,19,20,21,22,23,24,25,26,27,28,29,30,31,32,33,34,35,36,37,38,39,40,41,42,43,44,45]. Among these publications, 17 were focused on PT, 10 on photon RT, and 3 on a combination of photon RT and PBT. All the studies were retrospective except for one prospective study [21]. The details of these articles are shown in Table 1. Forest plots for each modality for 3- and 5-year OS are shown in Figure 2. A meta-analysis of the 30 articles found 3- and 5-year OS rates (PT vs. photon RT or combined photon RT/PBT) of 90.8% (95% CI: 87.4–93.3%) vs. 89.5% (95% CI: 83.0–93.6%), *p* = 0.6543; 80.0% (95% CI: 75.7–83.6%) vs. 89.5% (95% CI: 83.0–93.6%), *p* = 0.6787. In the meta-analysis, the 3-year PFS rate for PT group was 71.7% (95% CI: 63.1–78.6%). This rate could not be calculated for photon RT due to insufficient data. Forest plots for each modality for 5-year PFS are shown in Figure 3. The 5-year PFS rates (PT vs. photon RT or photon RT/PBT) were 67.8% (95% CI: 56.5–76.7%) vs. 40.2% (95% CI: 31.6–48.7%), *p* = 0.0004.

The random-effects meta-regression analysis with modality as an explanatory variable was adjusted primarily for age and gender. This was due to the high rates of missing data for other potential variables, such as GTR and tumor volume, which required exclusion of these variables from the analysis. The results are shown in Table 2. Neither age nor gender were significant factors for any of the indicators. Treatment modality (PT vs. photon RT or photon RT/PBT) was not a significant factor for 3-year OS (*p* = 0.42) and 5-year OS (*p* = 0.11), but was a significant factor for 5-year PFS (*p* < 0.0001). The rate of brain necrosis as a late adverse event is listed in Table 1 for each article. This rate was 8–50% in PT cases, but only 0–4% after photon RT.

## 4. Discussion

This study is the first meta-analysis comparing the efficacy of PT with photon RT for skull base chordoma. Comparisons of PT with photon RT for chordomas without specifying the site have been reported, but skull base chordoma needs to be treated independently due to the younger age of onset and low rate of complete resection [8]. The biological and physical properties of PT make it useful for skull base chordoma, but the rarity of the disease has prevented randomized comparative studies of PT and photon RT. The current study shows a significant benefit of PT over photon RT for skull base chordoma for 5-year PFS, and PT resulted in 3- and 5-year OS of 90.8% and 80.0%, and 3- and 5-year PFS of 71.7% and 67.8%. These findings show that the challenges posed by the rarity of the disease and the historical reliance on retrospective studies can be overcome using a meta-analysis.

The better PFS after PT compared to photon RT is thought to be due to the total dose and irradiation field. The median total doses in the analysis were 65–78.4 Gy relative biological effectiveness (RBE) for PT, and 14.8–81 Gy (RBE) for photon RT or photon RT/PBT [16,17,18,19,20,21,22,23,24,25,26,27,28,29,30,31,32,33,34,35,36,37,38,39,40,41,42,43,44,45]. Although there were differences in the dose per fraction, the total dose tended to be higher for PT. Additionally, there were differences in the field settings depending on the treatment modality. With Gamma Knife or linear accelerator-based stereotactic radiosurgery and stereotactic radiotherapy (SRS/SRT), the tumor margins are used to define the dose, with a higher dose delivered to the tumor center [35,36]. In contrast, PT planning frequently uses the pre-surgical extent of the tumor and setting of the clinical target volume (CTV) with a 5–10 mm margin to the gross tumor volume (GTV) [17,19,21]. Therefore, the better 5-year PFS with PT may be attributable to the higher total irradiation dose and broader field settings. It was also reported that the high PFS achieved by PT for chordomas is superior in terms of cost-effectiveness [46,47]. Generally, the cost of PT is higher than that of photon RT, but it implies the suppression of costs for reoperations and other procedures. While some reports indicate that PT may not offer significant cost-effectiveness for head and neck cancer due to similar treatment outcomes [48,49], it is considered cost-effective for treating skull base chordomas.

The efficacy of PT may be somewhat offset by the higher rate of brain necrosis after PT. Theoretically, the Bragg peak in PT allows for dose concentration, enabling high-dose delivery to the tumor while minimizing the dose to surrounding normal tissues, as required for skull base chordoma [50]. However, our latest retrospective analysis also identified a 13% occurrence of brain necrosis, with a high total dose found to be a risk factor for brain necrosis [51]. The volume of normal brain exposed to a high dose (over 60 Gy) is known to be a risk factor for brain necrosis, and setting a wider CTV for PT compared to photon RT may also contribute to the occurrence of brain necrosis [52]. In situations where the CTV is curved or U-shaped, intensity-modulated radiation therapy (IMRT) may be effective, and combined photon RT and PBT may also be useful [53]. Particularly for a U-shaped target area, using only passive PT might result in insufficient dose areas to spare the brain, raising concerns about reduced local efficacy [45]. However, this limitation can be overcome with the spot scanning method using intensity-modulated particle beam therapy (IMPT). Indeed, favorable treatment plans using IMPT for skull base chordoma can be created without compromising the dose to the tumor [27,54]. Evidence for use of IMPT is still not widely available, but this method does create superior dose distributions for many organs, not just skull base tumors, compared to IMRT or passive PT [55]. Therefore, as IMPT becomes more accessible, it is likely to facilitate further dose increases, leading to improved treatment outcomes.

The limitations of this study include the rarity of skull base chordoma and the small number of cases treated with photon RT. Due to this rarity, prospective studies are challenging to conduct, and almost all the papers used in this analysis were retrospective studies, which limits the quality of the analysis. In addition, information on dose fractionation, CTV settings, and GTR rates was missing in many papers, which made a thorough analysis difficult. Details of surgical treatment, which play a crucial role in the integrated treatment strategy for chordomas, significantly impact treatment outcomes. Future analyses, including the quality of surgery such as resection rates, are anticipated as results have improved with increased rates of complete resection in our past reports [26,31]. Additionally, performing a subgroup analysis by particle type to examine the differences in biological effects between proton and carbon ion therapy would be of great interest, but due to the lack of studies on carbon ion therapy, this analysis could not be conducted at this time. More detailed analyses will be possible as more treatment outcomes are published.

## 5. Conclusions

This study shows that particle beam therapy gives higher PFS compared to photon RT for skull base chordoma. However, there is also a tendency for a higher incidence of brain necrosis with particle beam therapy. Further studies are needed to validate these findings.

## Figures and Tables

**Figure 1 cancers-16-02569-f001:**
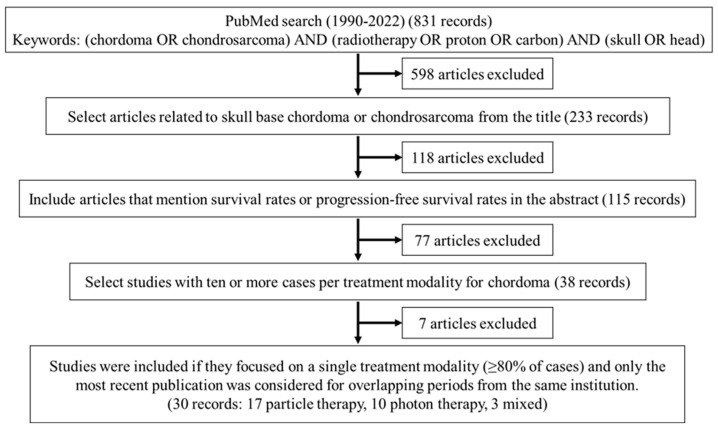
Flow diagram of study selection for systematic review and meta-analysis.

**Figure 2 cancers-16-02569-f002:**
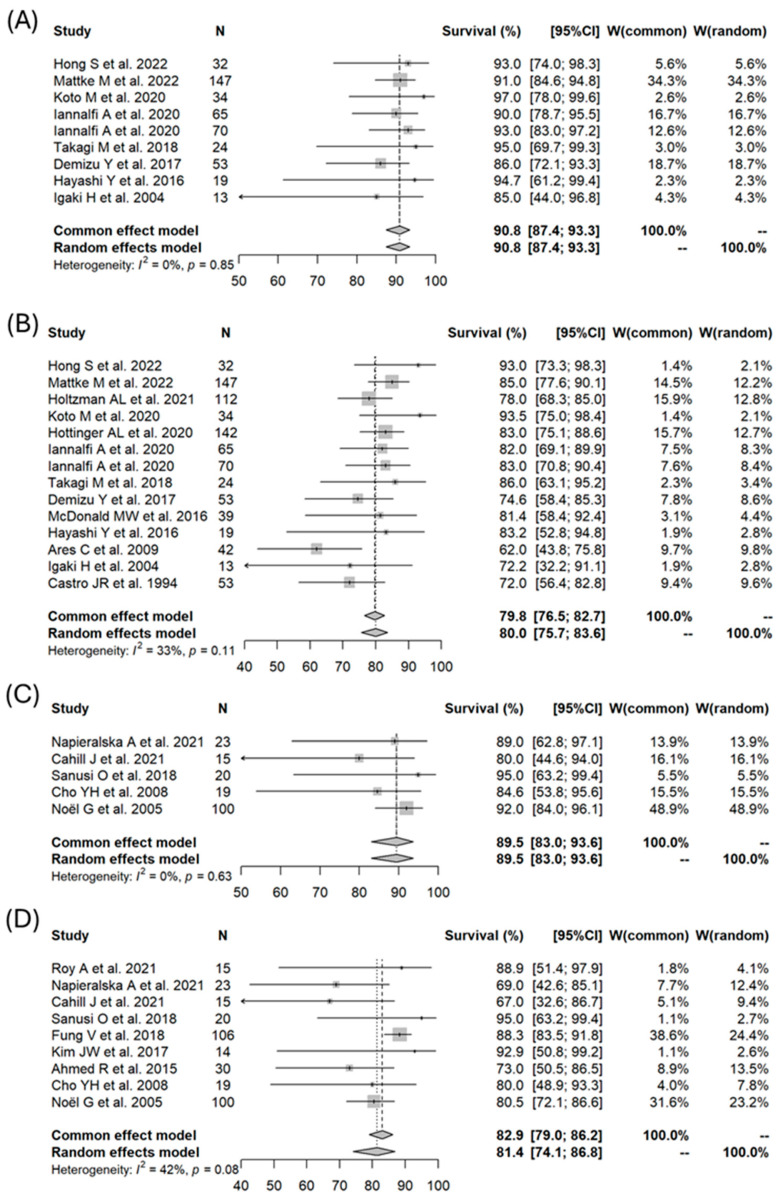
Forest plots of overall survival for skull base chordoma: comparison between particle beam therapy and photon radiotherapy. (**A**) 3-year OS for particle beam therapy [16,17,19,21,22,23,26,31], (**B**) 5-year OS for particle beam therapy [16,17,18,19,20,21,22,23,25,26,29,31,32], (**C**) 3-year OS for photon radiotherapy [34,35,37,43,44], (**D**) 5-year OS for photon radiotherapy [33,34,35,37,38,40,42,43,44].

**Figure 3 cancers-16-02569-f003:**
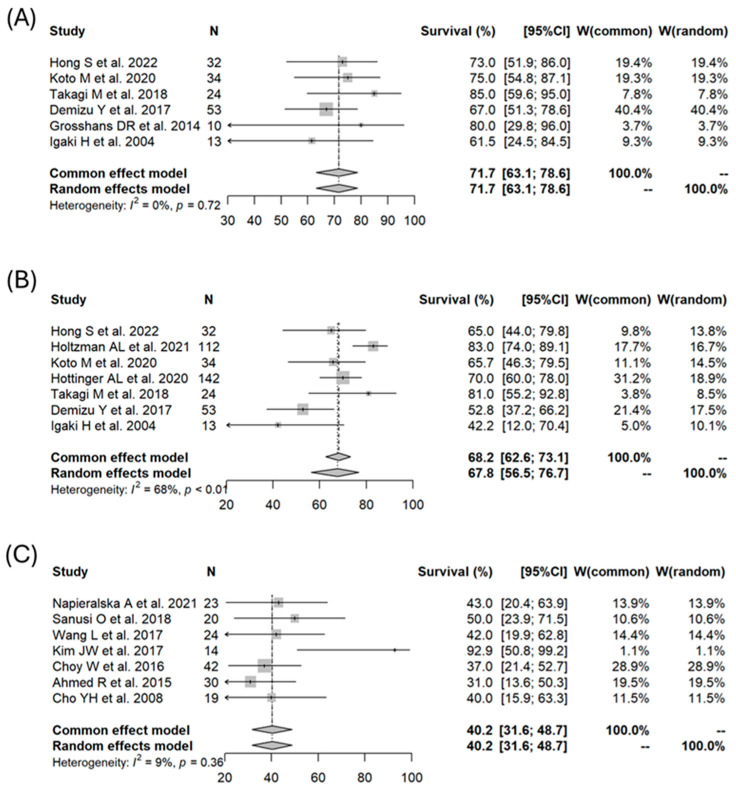
Forest plots of progression-free survival for skull base chordoma: comparison between particle beam therapy and photon radiotherapy. (**A**) 3-year PFS for particle beam therapy [16,19,22,23,27,31], (**B**) 5-year PFS for particle beam therapy [16,18,19,20,22,23,31], (**C**) 5-year PFS for photon radiotherapy [34,37,39,40,41,42,43].

**Table 1 cancers-16-02569-t001:** List of selected manuscripts.

Author	Year	Modality	*n*	Age (Median)	Male (%)	Tumor Volume (Median, cc)	GTR Rate(%)	Total Dose (Median, Gy (RBE))	3y OS (%)	5y OS (%)	5y PFS (%)	Brain Necrosis (%)
Hong [16]	2022	Proton	32	44	46.9		100.0	74	93	93	65	13
Mattke [17]	2022	Particle	147	51	57.8	40.4	0.0		91	85		14
Holtzman [18]	2021	Proton	112	52	68.8		97.3	73.8		78	83	4
Koto [19]	2020	Carbon	34	52	52.9	18.7	0.0	60.8	97	93.5	65.7	50
Hottinger [20]	2020	Proton	142	42	53.5	26.3	96.5	74		83	70	
Iannalfi [21]	2020	Carbon	65	58	64.6	13	0.0	70.4	90	82		30
Iannalfi [21]	2020	Proton	70	53	57.1	3.5	27.1	74	93	83		30
Takagi [22]	2018	Particle	24	56	41.7	17	0.0	65	95	86	81	8
Demizu [23]	2017	Proton	53	56				70	86	74.6	52.8	
Weber [24]	2016	Proton	151	43	57.0	35.4	100.0	74				
McDonald [25]	2016	Proton	39	52	53.8	24.5	12.8	77.4		81.4		18
Hayashi [26]	2016	Proton	19	52	42.1	19	42.1	78.4	94.7	83.2		16
Grosshans [27]	2014	Proton	10	43			100.0	69.8				20
Deraniyagala [28]	2014	Proton	33		78.8		93.9	74				
Ares [29]	2009	Proton	42		42.9			73.5		62		17
Weber [30]	2005	Proton	18	39		16.4		74				
Igaki [31]	2004	Proton	13	61	38.5	32.9	0.0	72	85	72.2	42.2	15
Castro [32]	1994	Particle	53	44	43.4		100.0	72	93	72		
Roy [33]	2021	GKRS	15					16		88.9		
Napieralska [34]	2021	SRS/SRT	23	53	52.2	17	21.7	52	89	69	43	
Cahill [35]	2021	GKRS	15	58	66.7	13	73.3	20	80	67		0
Hafez [36]	2019	GKRS	12	46	41.7	2.7	0.0	16				0
Sanusi [37]	2018	GKRS	20	47	55.0	23.07		14.8	95	95	50	0
Fung [38]	2018	Combined	106		56.6	25	4.8	73.8		88.3		4
Wang [39]	2017	GKRS	24	35	50.0	15.8	0.0	30.5			42	
Kim [40]	2017	IMRT	14	39	21.4		78.6	67		92.9	92.9	0
Choy [41]	2016	SRS/SRT	42	53	57.1	27.18	92.9	17.8			37	
Ahmed [42]	2015	RT	30	40	50.0			81		73	31	
Cho [43]	2008	RT	19	37	21.1		73.7	60.2	84.6	80	40	
Noel [44]	2005	Combined	100	53	60.0	23		67	92	80.5		1
Terahara [45]	1999	Combined	115	45	57.4	46		68.9		88.9		

GTR: gross total resection, RBE: relative biological effectiveness, GKRS: gamma knife radiosurgery, SRS: stereotactic radiosurgery, SRT: stereotactic radiotherapy, Combined: Combined photon radiotherapy and proton beam therapy, IMRT: intensity modulated radiation therapy, RT: radiotherapy.

**Table 2 cancers-16-02569-t002:** Meta-regression analysis of predictive factors for overall survival and progression-free survival.

Factors	Coefficient	SE	Lower CL	Upper CL	Z Value	*p* Value
3-year OS						
Modality	0.269	0.331	−0.379	0.918	0.814	0.416
Male ratio	−0.010	0.020	−0.049	0.029	−0.509	0.611
Age	0.025	0.041	−0.056	0.106	0.609	0.543
5-year OS						
Modality	0.300	0.187	−0.066	0.666	1.605	0.108
Male ratio	0.009	0.011	−0.012	0.030	0.819	0.413
Age	0.004	0.017	−0.030	0.039	0.251	0.802
3-year PFS						
Modality	0.441	2.713	−4.876	5.758	0.163	0.871
Male ratio	−0.007	0.076	−0.156	0.143	−0.086	0.932
Age	0.007	0.068	−0.126	0.140	0.105	0.916
5-year PFS						
Modality	0.944	0.216	0.521	1.367	4.372	<0.0001
Male ratio	−0.011	0.010	−0.031	0.009	−1.066	0.287
Age	0.008	0.016	−0.023	0.039	0.508	0.612

## Data Availability

The data supporting the findings of this study are available within the article. All data included in this systematic review and meta-analysis were derived from publicly available sources.

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
