# Peer review of "Systematic Review and Meta-Analysis of Particle Beam Therapy versus Photon Radiotherapy for Skull Base Chordoma: TRP-Chordoma 2024"

_cancers, 2024, doi:10.3390/cancers16142569_

Round 1

Reviewer 1 Report

Comments and Suggestions for Authors

Skull base chordoma is a radiation resistant tumor, and ion therapy, especially heavy ion therapy, has good therapeutic effects. Evidence-based medicine methods were applied in the manuscript  to study the efficacy of photon and ion therapy for skull base chordoma. The results of evidence-based medicine are closely related to the quality of the literature, but the article lacks literature quality evaluation analysis, and does not conduct subgroup analysis such as ion therapy type and whether surgery  performed.

Author Response

We appreciate the valuable comments and suggestions. we have revised the limitations section of the manuscript to address the impact of particle type differences, the effects of surgery, and the quality of the literature as follows. The changes have been highlighted in the text.

"The limitations of this study include the rarity of skull base chordoma and the small number of cases treated with photon RT. Due to this rarity, prospective studies are challenging to conduct, and almost all the papers used in this analysis were retrospective studies, which limits the quality of the analysis. In addition, information on dose fractionation, CTV settings, and GTR rates was missing in many papers, which made a thorough analysis difficult. Details of surgical treatment, which play a crucial role in the integrated treatment strategy for chordomas, significantly impact treatment outcomes. Future analyses, including the quality of surgery such as resection rates, are anticipated as results have improved with increased rates of complete resection in our past reports [26, 31]. Additionally, performing a subgroup analysis by particle type to examine the differences in biological effects between proton and carbon ion therapy would be of great interest, but due to the lack of studies on carbon ion therapy, this analysis could not be conducted at this time. More detailed analyses will be possible as more treatment outcomes are published."

Additionally, we have added the following sentence to the Results section:
"All the studies were retrospective except for one prospective study [21]."

Reviewer 2 Report

Comments and Suggestions for Authors

The analysis is very well conducted. Methods and results can be improved to underline which are the most significative items. Discussion also needs to be presented in differente form to underline the significance of the integrated strategy of chordomas. referral to surgery is fundamental as well as the complications and neurological complications. Finally the cost effectiveness is mandatory.

Comments on the Quality of English Language

There are minor errors; English naive speaker can improve the overall readability and some sentences need to be improved in form.

Author Response

We appreciate the valuable comments and suggestions. In response, we have revised the limitations section of the discussion regarding the integrated strategy for chordomas as follows. The changes have been highlighted in the text.

"The limitations of this study include the rarity of skull base chordoma and the small number of cases treated with photon RT. Due to this rarity, prospective studies are challenging to conduct, and almost all the papers used in this analysis were retrospective studies, which limits the quality of the analysis. In addition, information on dose fractionation, CTV settings, and GTR rates was missing in many papers, which made a thorough analysis difficult. Details of surgical treatment, which play a crucial role in the integrated treatment strategy for chordomas, significantly impact treatment outcomes. Future analyses, including the quality of surgery such as resection rates, are anticipated as results have improved with increased rates of complete resection in our past reports [26, 31]. Additionally, performing a subgroup analysis by particle type to examine the differences in biological effects between proton and carbon ion therapy would be of great interest, but due to the lack of studies on carbon ion therapy, this analysis could not be conducted at this time. More detailed analyses will be possible as more treatment outcomes are published."

Additionally, we have added the following discussion on cost-effectiveness:
"It has also been reported that the high PFS achieved by PT for chordomas is superior in terms of cost-effectiveness [46, 47]. Generally, the cost of PT is higher than that of photon RT, but it implies the suppression of costs for reoperations and other procedures. While some reports indicate that PT may not offer significant cost-effectiveness for head and neck cancer due to similar treatment outcomes [48, 49], it is considered cost-effective for treating skull base chordomas."